# Frequency-dependent selection can forecast evolution in *Streptococcus pneumoniae*

**Taj Azarian**[1,2]*‡, **Pamela P. Martinez**[2‡¤a], **Brian J. Arnold**[2], **Xueting Qiu**[2], **Lindsay R. Grant**[3], **Jukka Corander**[4,5,6], **Christophe Fraser**[7], **Nicholas J. Croucher**[8], **Laura L. Hammitt**[3], **Raymond Reid**[3], **Mathuram Santosham**[3], **Robert C. Weatherholtz**[3], **Stephen D. Bentley**[6], **Katherine L. O'Brien**[9], **Marc Lipsitch**[2,10‡], **William P. Hanage**[2‡]

**1** Burnett School of Biomedical Sciences, University of Central Florida, Orlando, Florida, United States of America, **2** Center for Communicable Disease Dynamics, Department of Epidemiology, T.H. Chan School of Public Health, Harvard University, Boston, Massachusetts, United States of America, **3** Center for American Indian Health, Johns Hopkins Bloomberg School of Public Health, Baltimore, Maryland, United States of America, **4** Helsinki Institute for Information Technology, Department of Mathematics and Statistics, University of Helsinki, Helsinki, Finland, **5** Department of Biostatistics, University of Oslo, Oslo, Norway, **6** Infection Genomics, The Wellcome Trust Sanger Institute, Wellcome Trust Genome Campus, Hinxton, United Kingdom, **7** Big Data Institute, Nuffield Department of Medicine, University of Oxford, Oxford, United Kingdom, **8** MRC Centre for Global Infectious Disease Analysis, Department of Infectious Disease Epidemiology, Imperial College London, London, United Kingdom, **9** World Health Organization, Geneva, Switzerland, **10** Department of Immunology and Infectious Diseases, T.H. Chan School of Public Health, Harvard University, Boston, Massachusetts, United States of America

¤a Current address: Department of Microbiology, University of Illinois at Urbana Champaign, Illinois, United States of America.
‡ TA and PPM are joint first authors on this work. ML and WPH are joint senior authors on this work.
* taj.azarian@ucf.edu

**Data Availability Statement:** Whole-genome sequencing data are available from NCBI under BioProject PRJEB8327: https://www.ncbi.nlm.nih.gov/bioproject/PRJEB8327. Accession numbers

## Abstract

Predicting how pathogen populations will change over time is challenging. Such has been the case with *Streptococcus pneumoniae*, an important human pathogen, and the pneumococcal conjugate vaccines (PCVs), which target only a fraction of the strains in the population. Here, we use the frequencies of accessory genes to predict changes in the pneumococcal population after vaccination, hypothesizing that these frequencies reflect negative frequency-dependent selection (NFDS) on the gene products. We find that the standardized predicted fitness of a strain, estimated by an NFDS-based model at the time the vaccine is introduced, enables us to predict whether the strain increases or decreases in prevalence following vaccination. Further, we are able to forecast the equilibrium post-vaccine population composition and assess the invasion capacity of emerging lineages. Overall, we provide a method for predicting the impact of an intervention on pneumococcal populations with potential application to other bacterial pathogens in which NFDS is a driving force.

## Introduction

Human interventions perturb microbial populations in many ways. Most obviously, the use of antibiotics or vaccines that target some strains and not others provide opportunities for new

and accompanying metadata have previously been published. All R code and associated input data frames as well as phylogenies are provided in the supporting information and are available on GitHub here: https://github.com/c2-d2/Projects/tree/master/NFDS.

**Funding:** TA and WPH were funded by NIH grant R01 AI106786. TA, PPM, and ML were funded by NIH grant R01 AI048935. XQ was supported by the National Institute of General Medical Sciences of NIH under Award Number U54GM088558. NJC was funded by a Sir Henry Dale fellowship and jointly funded by the Wellcome Trust and Royal Society (Grant Number 104169/Z/14/A). JC was funded by European Research Council grant number 742158. The funders had no role in study design, data collection and analysis, decision to publish, or preparation of the manuscript.

**Competing interests:** I have read the journal's policy and the authors of this manuscript have the following competing interests. ML has consulted for Pfizer, Affinivax, and Merck and has received grant support not related to this paper from Pfizer and PATH Vaccine Solutions. WPH, ML, and NJC have consulted for Antigen Discovery Inc. The authors have declared that no competing interests exist. KLOB has received grant support for pneumococcal work not related to this paper from Pfizer, GSK, and Gavi. KLOB has consulted for Merck and Sanofi Pasteur. LRG, LLH, and RCW have received grant support not related to this paper from Pfizer, Merck, and GSK.

**Abbreviations:** COG, clusters of orthologous groups; MGE, mobile genetic elements; NFDS, negative frequency-dependent selection; NVT, nonvaccine serotype; PCV, pneumococcal conjugate vaccine; PCV-13, 13-valent pneumococcal conjugate vaccine; PCV-7, 7-valent pneumococcal conjugate vaccine; SC, sequence cluster; VT, vaccine-serotype.

strains to emerge and become established. Examples include vaccines for antigenically diverse human pathogens like influenza, *Neisseria meningitidis*, *Haemophilus influenzae*, *Streptococcus pneumoniae*, and human papillomavirus [1–3]. Predicting these changes is a central goal of population genomic and evolutionary studies of pathogens as these data may inform vaccine development, surveillance activities, and drug discovery [4–7]. For bacteria in particular, detailed predictions of how a population will respond to a selective pressure are challenging. Models that specify how mutations with a given fitness change in frequency over time are often hard to apply in practice, as we typically do not know in advance important parameters such as the fitness value of particular alleles or how this is affected by their frequency (frequency-dependent selection) or genetic background (epistasis) [8,9].

Ongoing efforts to control disease caused by *S. pneumoniae* (the pneumococcus), a colonizer of the human nasopharynx and a cause of pneumonia, bacteremia, meningitis, and otitis media, underscore the difficulties of predicting changes after introduction of a vaccine [10]. Pneumococcal conjugate vaccines (PCVs) target only a fraction of this antigenically diverse species, which contains over 90 distinct serotypes defined by the antigenic properties of the polysaccharide capsule [11]. In the United States, the 7-valent PCV7 vaccine was introduced in 2000 and targeted 7 serotypes (4, 6B, 9V, 14, 18C, 19F, and 23F) that were responsible for the greatest proportion of invasive pneumococcal disease, whereas the successor 13-valent pneumococcal conjugate vaccine (PCV13), was introduced in 2010 and targeted 13 serotypes (1, 3, 4, 5, 6A, 6B, 7F, 9V, 14, 18C, 19A, 19F, and 23F) [12]. Following widespread introduction of PCVs among children, the carriage prevalence of pneumococci remained largely unchanged (approximately 30% of the general US population) [13,14] and nonvaccine serotypes (NVTs) benefited from the removal of their vaccine-serotype (VT) competitors and became more common in carriage and disease, with the gains from reducing VT disease partly offset by increases in NVT disease [14–16]. In some instances, this led to an increase in NVT serotypes capable of causing severe disease or that harbored genes conferring antimicrobial resistance [17,18]. These changes in the pathogen population varied by location and were not fully appreciated until retrospective analysis [19–21].

Our recent study of pneumococcal carriage isolates that were collected before and after PCV7 vaccine introduction in the Southwest US [21] illustrates the complexity in post-vaccine population dynamics, echoing findings from other studies. Pneumococcal populations contain multiple "sequence clusters," which are closely related lineages, defined on the basis of sequence variation in loci present among all isolates (i.e., the core genome) [22]. We henceforth use the term *strains* to refer to these lineages/sequence clusters and note that different isolates of a single strain may exhibit different serotypes. In general, although their frequencies vary markedly between locations, the dominant *S. pneumoniae* strains are widespread [23,24]. Variation in genome content due to horizontal gene transfer is a hallmark of prokaryotes; therefore, in addition to the core genome, we can define the accessory genome, as those genes not found in all isolates in the sample [25,26]. Consistent with their close phylogenetic relatedness in terms of core genome sequence variation, each strain we identify is composed of isolates that are fairly homogeneous—but not completely so—in the presence/absence of accessory genes as well as phenotypic properties such as serotype and antibiotic resistance [27].

Previous work showed that post-vaccine success of pneumococcal strains may depend on the accessory genome [24,28]. In many bacteria, this can be a large fraction of the total number of genes found in a species (i.e., the "pangenome") [29,30]. A population genomic study of pneumococci in Massachusetts children found that vaccination had remarkably little effect, after 6 years, on the overall frequencies of individual accessory genes (defined as clusters of orthologous groups [COGs]) [28]. Despite the fact that nearly half the pre-vaccine population

had serotypes targeted by the vaccine, only 2 of >3,000 loci in the accessory genome significantly decreased in frequency 6 years post-vaccine, and none increased [28]. More recently, a geographically diverse sample of pneumococcal genomes showed that although the distribution of strains varied widely across the globe, the proportion of isolates in each sample containing each individual accessory gene was highly consistent across locations [24]. Where vaccine was introduced, accessory gene frequencies were perturbed by the removal of vaccine types but trended back toward their pre-vaccine frequencies over time [21,24,31]. Negative frequency-dependent selection (NFDS) was proposed as the mechanism by which the frequencies of loci were restored after vaccine introduction [24]. NFDS is a type of balancing selection, which maintains diversity by favoring variants when rare but exacting a cost when they become common, such that the frequency of the variant stabilizes at intermediate values or, in some instances, result in frequency oscillations [9]. Examples of mechanisms produced by NFDS include host immunity and bacteriophage predation, and as such, balancing selection is recognized as a key contributor to population composition and diversity [32,33]. Among pneumococci, similar processes have been proposed to explain the coexistence of multiple serotypes [34] and vaccine-induced metabolic shifts [35].

Here, we present flexible, easily computable statistics that estimate the fitness of any strain using the contents of its accessory genome as a proxy for how it will be affected by NFDS, dependent on the frequencies of other strains in the population and specifically, of the accessory genes they carry. Even though we do not know the specific loci under selection or the mechanism involved, we are able to make predictions about the composition of a population as well as predict the fitness of any strain in any population, whether or not it has yet appeared in that population. Overall, this predictive model offers a way to study population processes and the response to interventions.

## Results

In the sample of 937 pneumococcal isolates composed of 35 strains from the Southwest US, we observed a sharp decline in PCV7-VT strains following vaccination (Fig 1 and S1 Fig). VT strains were subsequently replaced by NVT strains, including 2 emergent NVT strains that had not been observed pre-vaccine, although they were present during the same time period in a carriage dataset from Massachusetts [21,28]. We first show that there was considerable deviation from the null expectation that NVT strains would increase in prevalence pro rata to their pre-vaccine frequency; the most common NVT strains before vaccination were not necessarily the most prevalent 12 years after the introduction of the vaccine (Fig 1A). In particular, we find 13 of 35 strains deviated significantly from the prevalence expected under a null pro rata model; 9 were more common than expected, and 4 were less common (annotated with plus and minus signs, respectively, in Fig 1B). The impact of vaccination on individual NVT strains was, hence, not easily predictable. Consequently, public health authorities and vaccine manufacturers have had to rely on surveillance after the introduction of PCV7 to estimate the next epidemiologically important lineage and determine subsequent vaccine formulations. At best, this uncertainty reduces the population impact of vaccination; at worst, it could unintentionally increase the prevalence of virulent or antibiotic resistant lineages [36].

Having documented that there were strains that increased significantly more or less than their pre-vaccine frequency would indicate, we sought to define a parsimonious predictive algorithm based on NFDS that could account for these changes. We hypothesized that evolutionary dynamics could be predicted on the premise that after perturbation by vaccine, strains characterized by accessory genomes that could best restore the pre-perturbation accessory gene frequency equilibrium would have the highest fitness and therefore increase in prevalence

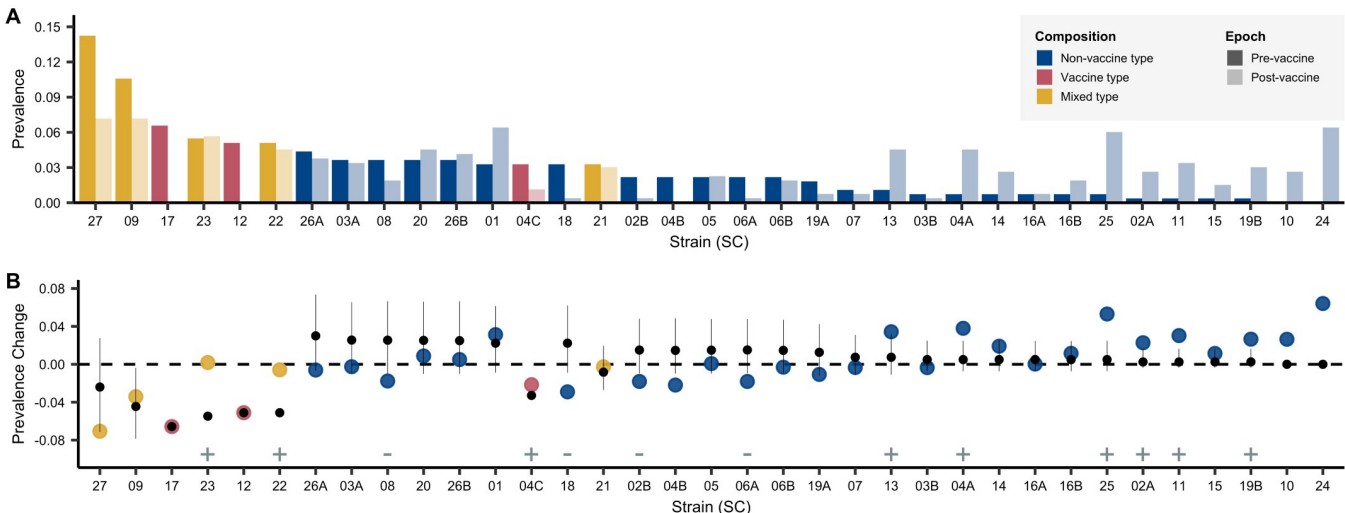

**Fig 1. Strain's prevalence.** A. Pre-vaccine to post-vaccine SCs. Strains are ordered from highest to lowest pre-vaccine prevalence. Raw data used in this figure are included in S1 Table. Observed prevalence change calculated as post-vaccine frequencies minus pre-vaccine frequencies. Changes in prevalence are compared with that expected under a pro rata null model (i.e., not using the predictive methods in this paper). Observed changes in prevalence are represented by points colored by the serotype composition of the strain: NVT only, PCV7 VT only, and VT-NVT. The point and whiskers show the prevalence change expected if all VT strains were removed and NVT increased proportional to their pre-vaccine prevalence—i.e., in a null model of pro rata increase in which only the VT strains were removed and all NVT strains increased equally in proportion to their pre-vaccine prevalence. The dot is the median, and the whiskers give the 2.5% and 97.5% quantiles of predicted changes under the null model using 10,000 bootstraps from pre-vaccine samples. Significant differences between the changes in prevalence from the pro rata model and the observed data are denoted with plus and minus signs specifying strains that were significantly more ($n = 9$) or less ($n = 4$) common, respectively. Among the most successful were strains that contained both VT and NVT isolates (SC-22 and SC-23) whose NVT component included serotypes 6C, 15C, and 35B, as well as SC-24 and SC-25, which were dominated by the NVT serotypes 23A and 15C, respectively. SC-27 is polyphyletic, composed of an aggregate of strains that are at low frequency in the overall population. Compared with strains composed of solely NVT isolates, those with mixed NVT-VT had marginally higher risk differences, indicating greater success than expected under the null model ($\beta = 0.03$, SE = 0.015, F(1,29) = 3.67, $p = 0.06$). Two strains that emerged during the study period (SC-10 and SC-24) were not included in this analysis as they were not present at the first time point. See S1 Data and S1 Code for details. NVT, nonvaccine serotype; SC, sequence cluster; SE, standard error; VT, vaccine serotype; VT-NVT, mixed VT and NVT.

disproportionately. To this end, we implemented a deterministic model using the replicator equation to calculate the fitness of a strain based on its accessory genome, using vaccination as an example of perturbation [37–39] (Eq 1).

$$\frac{dx_i}{dt} = x_i(\omega_i - \varphi), \quad \varphi = \sum_{j=1}^{n} x_j \omega_j \tag{1}$$

Under this formulation, $x_i$ denotes the frequency of strain $i$ ($i = \{1, \ldots, n\}$) after the vaccine introduction, $n$ is the total number of strains, $\omega_i$ denotes the fitness of strain $i$ (adapted from Ref. [24]), and $\varphi$ is the average population fitness. The difference ($\omega - \varphi$) is a standardized predicted fitness. The fitness vector $\omega$ is defined as the product of 2 elements (Eq 2), matrix **K** and the vector ($e - f$). The element $k_{i,l}$ of matrix **K** is a value between 0 and 1 and refers to the frequency of accessory gene $l$ in strain $i$. The element $l^{th}$ of the vector ($e - f$) corresponds to the difference between accessory gene $l$'s pre-vaccine frequency and its post-vaccine frequency. Here, $f_l$ is defined as the frequency of gene $l$ among the isolates that are nonvaccine types. Intuitively, this vector represents the gap that vaccination produces in the population in terms of the accessory loci it removes, and $\omega_i$ quantifies the ability of strain $i$ to fill that gap. In contrast with previous work [24], we do not define carrying capacity or migration rates, requiring only knowledge of the accessory gene frequencies at equilibrium and which strains they are associated with; these quantities can be estimated from a population survey prior to the perturbation of interest. We assume that the impact of recombination on the accessory genome is negligible

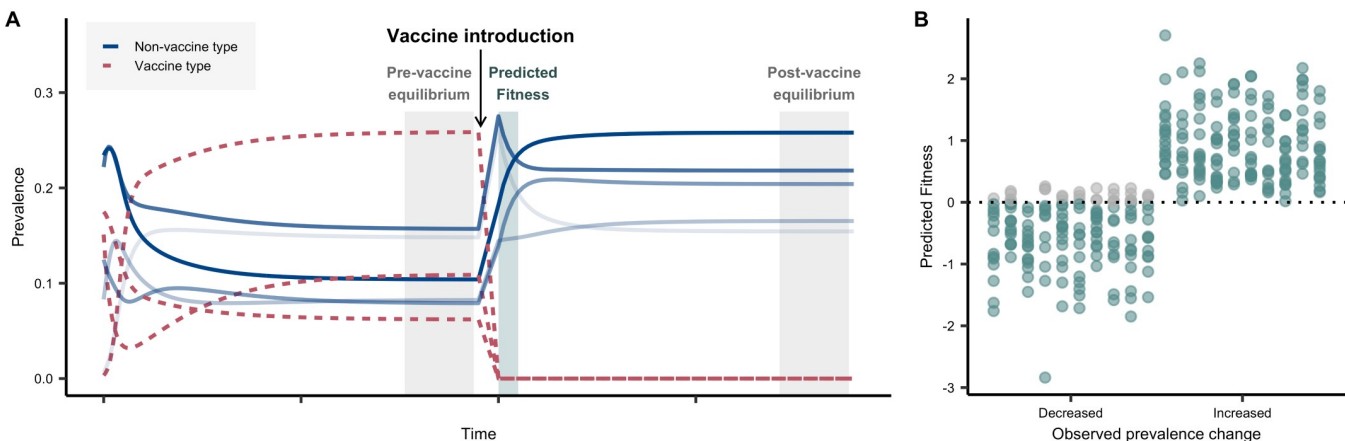

**Fig 2. Simulations.** A. Conceptual diagram for simulations. Descriptive representation of the strain prevalence at different stages relative to vaccine introduction: pre-vaccine equilibrium, vaccine introduction, and post-vaccine equilibrium. We modeled a population of VT and NVT strains (represented as unique genotypes with alleles 1 or 0 at a locus, denoting the presence or absence of a single accessory locus) and simulated the removal of VT genotypes, following the post-vaccine population to equilibrium (details in methods). In this illustrative figure, 8 strains are shown, with their prevalence in the population evolving over time. The system is allowed to evolve until it reaches a steady state ("pre-vaccine equilibrium"). Three strains were then targeted to mimic a vaccine introduction, which removes them from the system. The predicted fitness was then estimated from the period just after the vaccine introduction, when the population has been depleted of VT but relative prevalence of NVT has not changed—a quantity that can be calculated from pre-vaccine data alone. Finally, the system reaches a second steady state ("post-vaccine equilibrium"). Different shades of blue represent the rank of the strain frequencies in the post-vaccine equilibrium. B. Predicted fitness. Comparison of the direction of prevalence change of strains from pre- to post-vaccine using simulated data and predicted fitness from these simulated data. For these 10 replicate simulations, 2,371 accessory loci and 35 randomly chosen strains were simulated, including 3 VT genotypes. For each replicate, the pre-vaccine equilibrium frequencies of the 2,371 accessory loci were varied. Final prevalence of strains was obtained by quadratic programing, and prevalence change for each NVT strain was calculated as post-vaccine prevalence minus pre-vaccine prevalence, in both cases with all NVT strains summing to 100%. Each column in the decreased and increased category represents the results from 1 simulation (i.e., the first column in the decreased category corresponds to the first column in the increased category, and the dots sum to 32). The predicted fitness of the strain accurately predicts the direction of the prevalence change in 92.8% of cases (teal dots). Gray dots represent instances in which the direction of the prevalence change was not predicted correctly in the simulation. See S1 Data and S1 Code for details. NVT, nonvaccine serotype; VT, vaccine serotype.

over the time period we study here.

$$\omega \;=\; \mathbf{K}(e - f) \tag{2}$$

Using simulated data, we first assessed the ability of a strain's standardized predicted fitness ($\omega - \varphi$) (for brevity, we drop the modifier "standardized" hereafter) to predict the direction of its change in frequency, based on its ability to resolve the vaccine-induced perturbation (Fig 2). Note that this predicted fitness uses only data available before vaccine rollout. Using this model, we show that in simulated datasets, the predicted fitness is consistent with the direction of a simulated strain's adjusted prevalence change (i.e., changes in prevalence minus what would be expected if all NVT strains increased by the same proportion from their pre-vaccine prevalence) 92.8% of cases (Fig 2B).

Next, we asked whether this approach could predict the post-vaccine composition of an actual pneumococcal population and specifically, the relative contribution of each strain to serotype replacement. For each strain present before vaccine introduction, we used the accessory genome to calculate the strain's fitness following the removal of vaccine types. We identified 2,371 genes that were present in between 5% and 95% of isolates. In this dataset, we found the predicted fitness value was significantly and positively correlated with the observed prevalence change (Adjusted $R^2 = 0.41$, $p \ll 0.001$, Fig 3A). Further, the trajectory following vaccination, whether increasing or decreasing in frequency, was accurately predicted for 28 of the 31 tested strains identified in the sample, as indicated by the upper right and lower left quadrants of Fig 3A. Strains with a positive prevalence change had substantially higher predicted

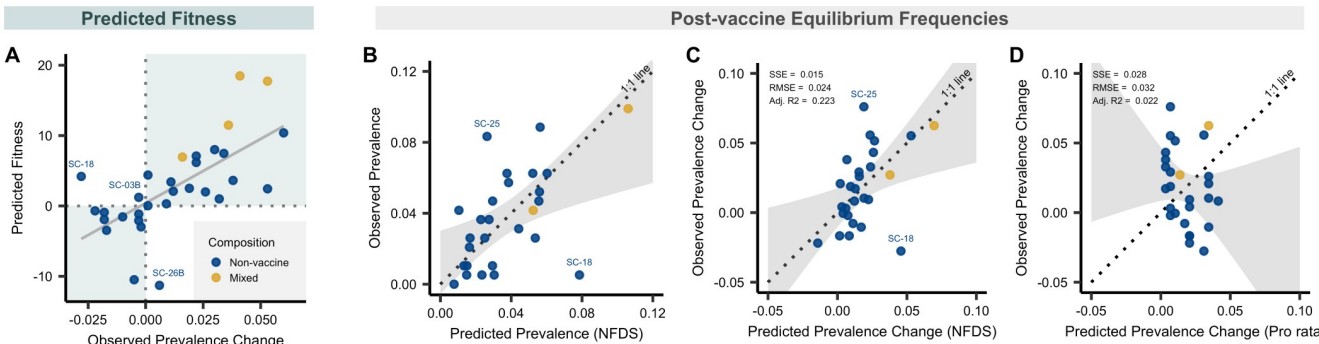

**Fig 3. Predicted fitness and predicted prevalence. A**, Relationship between predicted fitness and observed prevalence change from pre- to post-vaccine among 31 strains, in each case summing to 100%. Prevalence change was calculated as post-vaccine frequencies minus pre-vaccine frequencies. Predicted fitness was calculated using data solely from the pre-vaccine sample ($n = 27$), with the exceptions of strains for which there were no NVT isolates present in the sample before the introduction of PCV7 ($n = 4$). For those strains, data were imputed from the time point during which they were first observed. Four strains were excluded either because they were polyphyletic (SC-27) or had no NVT isolates present pre- or post-vaccine, and therefore could not be imputed (SC-04C, SC-12, and SC-17). The points are colored by serotype composition of strains: nonvaccine types in blue and mixed VT and NVT types in yellow. The shaded quadrants indicate regions of accurate prediction of the prevalence change direction (increased post-vaccine versus decreased) given the predicted fitness value. Three outlier strains are annotated for which the predicted direction of their prevalence change differed from that which was observed (i.e., they were predicted to increase based on their fitness when their prevalence from pre- to post-vaccine decreased, or vice versa). **B**, Scatterplot of observed versus predicted prevalence of 27 strains at post-vaccine equilibrium based on quadratic programming. These 27 strains contained at least 1 NVT strain pre-vaccine. Points are colored based on serotype composition as described in panel **A**. Perfect predictions would lie on the dotted line of equality (1:1 line). The shaded gray region shows the CI from the linear regression model used to test for deviation of the observed versus predicted values compared with the 1:1 line. Two outliers are annotated for which the difference between their predicted and observed prevalence was >1.5 times the interquartile range of the distribution of predicted and observed prevalence differences. As a note, the predictions remained significant if SC-09 (the extreme strain at 10% prevalence in B) was removed (slope, 95% CI 0.021–1.05; intercept, 95% CI −0.003 to 0.03; $p = 0.19$, chi-squared = 3.5). **C-D**, Comparison of the predicted prevalence change from quadratic programming analysis using accessory genes and naive pro rata model as shown in Fig 1B but applied to just these 27 strains. The dotted line of equality (1:1 line) and CI (gray) are shown as in panel B. Goodness-of-fit statistics including SSE, RMSE, and degrees of freedom Adj. $R^2$ are given for each model. The lower SSE and RMSE indicate a better model fit. See S1 Data and S1 Code for details. Adj. $R^2$, adjusted *R*-squared; NFDS, negative frequency-dependent selection; NVT, nonvaccine serotype; PCV7, 7-valent pneumococcal conjugate vaccine; RMSE, root mean squared error; SC, sequence cluster; SSE, sum of squared errors; VT, vaccine serotype.

fitness than those with a negative one (mean fitness of strains that increased versus decreased 6.4 versus −2.4; 95% CI 5.0–12.5, $p < 0.001$).

Although the predicted fitness estimates how successful each strain will be immediately following vaccination, the long-term post-vaccine prevalence or change in prevalence of each strain is of more direct interest for evolution and public health. Thus, we posited that over time post-vaccine, accessory gene frequencies would evolve to match as closely as possible to match those present pre-vaccine, and we used an optimization technique, quadratic programming, to calculate the NVT strain composition that produced these gene frequencies closest to those observed in the pre-vaccine population. Here, we specifically focused on only the 27 strains that were observed pre-vaccine in the Southwest US sample, allowing a projection with only data that were available at the time of vaccine introduction. This approach predicted the strain composition of the population following vaccination well, characterized by a 95% CI of the observed versus predicted post-vaccine strain frequencies that includes the line of equality (1:1 line), which denotes a perfect prediction, and by an intercept and slope that does not differ significantly from zero and one, respectively ($p = 0.24$; intercept 95% CI −0.005 to 0.030; slope 95% CI 0.257–1.075, Fig 3B). Similar results were obtained when comparing predicted and observed change in prevalence (Fig 3C), where again the dotted line of equality fell within the 95% CI of the regression of observed versus predicted change in prevalence ($p = 0.75$; intercept 95% CI −0.02 to 0.01; slope 95% CI 0.23–1.36). In comparison, a naive pro rata estimate based solely on pre-vaccine prevalence performed poorly in predicting the prevalence change (Fig 3D, $p = 0.001$; intercept 95% CI −0.05 to −0.008; slope 95% CI −1.43 to 0.35).

**Table 1. Comparison of pre- to post-vaccine prevalence change predictions using multiple models.**

| Model | $n_{loci}$ | Adj. $R^2$ | SSE | RMSE |
|---|---|---|---|---|
| Pro rata (proportional change) | NA | 0.022 | 0.028 | 0.032 |
| Accessory genome (NFDS) | 2,371 | 0.223 | 0.015 | 0.024 |
| Accessory genome (NFDS)—Sensitivity analysis (2010 only) | 2,371 | 0.081 | 0.024 | 0.030 |
| Core genome (NFDS) | 17,101 | 0.173 | 0.016 | 0.024 |
| Metabolic loci (NFDS) | 5,853 | 0.154 | 0.017 | 0.025 |

Goodness-of-fit statistics including SSE, RMSE, and degrees of freedom Adj. $R^2$ are given for each model in relation to the 1:1 line. Fit statistics are provided for the naive pro rata model and quadratic programming models using accessory genes, 5,853 biallelic polymorphic nucleotide sites found in 272 core genome metabolic genes, and 17,101 biallelic polymorphic nucleotide sites found in 1,111 core genes. The results of the sensitivity analysis using a subsample of 119 isolates collected in 2010 prior to the initiation of PCV13 vaccine introduction is also presented for the accessory.

Adj. $R^2$, adjusted $R$-squared; NFDS, negative frequency-dependent selection; PCV13, 13-valent pneumococcal conjugate vaccine; RMSE, root mean squared error; SSE, sum of squares due to error.

In the Southwest US, the 13-valent PCV13 vaccine was introduced during the second half of our post-vaccine sampling [21]. Despite this, the prevalence of PCV13 vaccine serotypes remained largely unchanged, suggesting little impact of this vaccine over the period of our study. To test the potential effect on our current analysis, we partitioned the post-vaccine sample into pre- and peri-PCV13 (i.e., around the time that PCV13 was initially introduced but before a significant proportion of the population was immunized), and the results are provided in Table 1, which demonstrate that our predictions were robust to subsampling. Finally, we tested the predictive value of different genomic elements, which are linked to accessory genes, finding that core genome loci ($n_{loci}$ = 17,101) and metabolic loci ($n_{loci}$ = 5,853) were also capable of predicting the impact of vaccine, though not as accurately as the accessory genome based on goodness-of-fit statistics (Table 1). This finding must be considered in the context of recombination, selection, and the evolutionary timescale impacting the pneumococcal genome, which may impact the varying magnitude of NFDS signal across sets of loci. Despite moderate levels of bacterial recombination among pneumococci, there remains appreciable linkage disequilibrium between loci nearby as well as genome-wide [8], which makes it difficult to discern the relative selective importance of any particular locus. Exactly which genomic elements are responsible for the predictive ability we document here is unknown but is obviously of interest and should be a focus for future work.

To further support our findings, we examined a previously published carriage dataset of pneumococci colonizing children in Massachusetts. This dataset is imperfect in several respects. First, it was smaller ($N$ = 616), particularly the initial sample from the population, which had only 131 isolates and came in the first year of vaccine introduction rather than before it. Also making this dataset less ideal, the last sample was obtained only 6 years after the first sample, giving less time for evolution to occur than in our Southwest US dataset. Changes in strain frequencies are shown in S2A and S2B Fig. Despite the limitations of the dataset, applying the same quadratic programming approach, we could predict the post-vaccine equilibrium prevalence of the 9 strains used in the analysis ($p$ = 0.65; intercept 95% CI −0.05 to 0.09; slope 95% CI 0.25–1.33) better than the pro rata model (S2C–S2E Fig).

Last, by combining the Massachusetts and Southwest US datasets [28,40], we assessed the capacity for migration and emergence of the 2 strains that were first observed in the Southwest US sample after the introduction of the PCV7 vaccine, shown as SC-10 (serotype 19A; ST320) and SC-24 (serotypes 15A, 23A/B; ST338) in Fig 1 and S1 Fig. These 2 strains were present at

the initial time point in the Massachusetts dataset (2001–2004) and also increased in prevalence thereafter. We found that they have a predicted fitness of 8.6 and 7.2, for SC-10 and SC-24 respectively, higher than any of the other potential migrant strains that were not present in our Southwest US sample before vaccination, indicating that their accessory gene frequencies were well adapted to offset the PCV7 perturbation in the Southwest US population. Indeed, only 2 of the strains present before vaccination in the Southwest US (SC-23 and SC-9) had a higher predicted fitness, suggesting that this approach can be used to quantify which strains are most likely to successfully invade a population.

## Discussion

We show that by estimating the fitness of strains using an NFDS-based model and the frequencies of accessory genes, we are able to predict the direction of prevalence change following vaccination and more broadly, the post-vaccine population composition. The ability of this type of balancing selection to determine the strain composition of a population is consistent with findings from environmental microbiology on multiple bacterial species [33]. Among pneumococci, changes in population dynamics after the introduction of vaccine have been explained by selection on many different aspects of the organism, including metabolic types, antibiotic resistance, carriage duration, recombination rates, and serotype competition, all of which are likely to be relevant contributors alongside, or components of, the accessory genome [35,36,41,42]. We provide a simple and effective approach for estimating the fitness of any strain in a population evolving under NFDS acting on accessory loci. All that is required is knowledge of the strain composition of the population and the accessory loci associated with each strain, as this approach does not depend on NFDS acting on particular known biological functions to predict the consequences of vaccination. It is quite conceivable that a minority of loci are involved, including even SNPs in the core genome, which also show a correlation (see Table 1 and [24]). We do not wish to imply that the sorts of selection discussed here act alone. Our previous work suggests the interplay between host immunity and polymorphic protein antigens may play a significant role [43], and other work suggests an important role for metabolic loci in the core genome [35]. Phage predation and defense as well as antibiotic resistance all likely contribute to the observed signal [27].

The pneumococcus is well known for being naturally transformable and indeed, exhibits high rates of homologous and nonhomologous recombination, which can vary quite markedly among strains [27,44]. Recombination plays a significant role in the movement and therefore distribution of accessory loci as well as the evolution of new strains, which, together with mutation, dictates population structure. However, the timescale over which recombination results in measurable changes to a population is unclear. In some instances, dramatic changes may occur rapidly, such as capsular switches that lead to the emergence of new strains [45]. In others, such as the movement of large mobile elements, the process may occur much slowly, as those elements have been found to be relatively stable within strains [27]. Of note, the strains included in the present study are estimated to have emerged, on average, 90 years ago, with the youngest strain (SC10) emerging around the introduction of PCV7 in the US (2010) [21]. Despite this seemingly long duration, the distribution of accessory loci within a strain is relatively uniform. Further, although recombination makes conventional phylogenetic analysis problematic, the methods we use to define strains [46] are not phylogenetic in nature, and we are not concerned with the details of the relationships between or within the strains, merely the definition of strains as a lineage of closely related bacteria that share characteristics of their core and accessory genomes.

There are 2 primary ways in which NVT strains can fill the gap left in the population by vaccination, depending on their genomic relatedness to the removed PCV7 VT strains. First, NVT taxa that are closely related to VT strains in core and accessory genomes are opportune replacements and are therefore expected to be more successful than average following vaccination. There are 2 strains in our dataset that are examples of this, which both increased after vaccination (see SC-09 and SC-23 in Fig 1 and S1 Fig). We therefore expect that for any strain that contains both VT and NVT representatives, the NVT fraction will increase post-vaccine, especially because these NVT taxa are sometimes similar to their VT counterparts in terms of serotype properties such as capsule thickness and charge, which are independently correlated with prevalence [47,48]. A good example of this is the serotype 15B/C component of strain SC-26 of the Southwest US sample, which we now predict to be successful following the more recent introduction of a vaccine incorporating 6 additional serotypes (PCV13) and which has indeed been noted to be increasing in certain locations [49–51]. Second, where such close relatives are not available, the pre-vaccine frequencies of accessory genes can be restored by other NVT that are divergent in core genomes but similar in accessory genomes. This association likely often results from the movement of mobile genetic elements (MGEs) in the population (e.g., phages and transposons) or nonhomologous recombination, which can make distantly related strains more similar in terms of genome content. As illustrated by the pairwise comparison of core/accessory genome divergence and absolute fitness difference of each strain (S3 Fig), there is an appreciable range of differences in fitness for strains that are equidistant in core and accessory genome divergence.

Certainly, as shown by outliers to predictions in Fig 3, we acknowledge that the model does not currently capture all population dynamics. Variation among loci in the strength of NFDS could account for some of these discrepancies, as indicated by retrospective model fitting. Other explanations include differences in the distribution of antibiotic resistance genes or possible vaccine cross-reactivity. For example, SC-18, containing serotype 6C, declined despite a positive predicted fitness; however, cross-reactivity between the PCV7 6B vaccine component and 6C may in part explain this observation [52]. Last, as components of the accessory genome may be redundant, meaning they encode similar functions or represent a single mobile element, our predictive power may be limited. Nevertheless, given the many potential pressures, mostly not directly observable, that we might expect to structure the pneumococcal population, it is notable how effectively this approach can predict the impact of this perturbation. Overall, we find a significant relationship between predicted fitness and the adjusted prevalence change of a strain. By optimizing the prevalence of each strain conditional on the gene frequencies before vaccination, we can estimate the equilibrium population after vaccination, using both the predicted fitness and numerical approximations of the post-vaccine equilibrium.

Predicting evolution is a central goal of population genomics especially when related to pathogens and human health. Although evolutionary theory provides an understanding of bacterial population processes including the relative success of lineages, distribution of phenotypes, and ecological niche adaption, these analyses are often conducted retrospectively. Here, we demonstrate a method for predicting the impact of perturbing the pneumococcal population that may be useful to predict the outcomes of future interventions including vaccines. An example application is the ability to predict the increase in strains 10 and 26, which are largely composed of serotype 19A and are significant for their antibiotic resistance and virulence [18]. The observed increase in invasive disease caused by serotype 19A after the introduction of PCV7 led to its later inclusion in PCV13. In principle, predicting the increase of 19A prior to PCV7 development would have guided decisions on serotype composition of the vaccine. Further, by incorporating information on invasive capacity based on serotype and epidemiological

data, these predictions could be extended to inform changes in invasive disease rates [53]. Overall, these dynamics suggest novel vaccine strategies in which one could target those strains whose removal would result in a predicted re-equilibration that favors the least virulent or most drug-susceptible lineages [54].

Our findings suggest numerous potential directions for future work, among them identifying the specific accessory loci or other genomic elements that are responsible for what we observe. Expanding the model to include immigration of other strains and disentangling the relative contribution of selection on various loci is likely to be a fruitful area for future research. One area worth exploring is the degree to which recombination acts to maintain gene frequencies on the timescale of population-level shifts in lineage composition. The emergence of new strains, characterized by novel combinations of accessory loci, is expected to be limited by the other strains present in the population in ways that are currently not well understood. The pervasive finding of accessory genomes in most bacterial species is usually explained by specialization of lineages to specific niches; however, it could also reflect widespread NFDS, and so future work should seek for evidence of similar signal in the core and accessory genome of other bacteria [55].

## Methods

### Study population and descriptive statistics

The Southwest US dataset used in this study is a subset of 3 studies of pneumococcal carriage conducted among Native American communities in the Southwest US from 1998 to 2012, as previously described [13,56,57]. This genomic dataset was particularly selected because (1) it includes the earliest cross-sectional sample of *S. pneumoniae* in a defined population, and it is the only sample that predates the introduction of PCV7; (2) the study period, 15 years, is the longest duration of observation of any genomic sample; (3) as a site for PCV7 and PCV13 vaccine trials, the population is well characterized, and the epidemiology of pneumococcal disease well understood; and (4) it is focused on isolates from carriage and not invasive pneumococcal disease—isolates from invasive disease only capture a sub-sample of the pneumococcal population because some strains are more invasive than others.

The pre-vaccine sample was collected from the well-defined control communities of the group-randomized trial of the PCV7 vaccine [57]. Pre-vaccine isolates included in our study were collected between March 1998 and April 2001. In late October 2000, PCV7 vaccination became routine, including catch-up for children aged <5 years. By March 2001, a total of 88% of 3- to 4-month-old infants living in PCV7-randomized communities and 77% of those in control communities had received >1 dose of PCV7 [58]. However, only 7 of the 274 isolates in our pre-vaccine sample were collected between October 2000 and March 2001; therefore, we feel it is reasonable to treat it as a pre-vaccine sample from an unperturbed population. The PCV13 was later introduced in 2010. The pneumococcal sample was subdivided into 35 sequence clusters (SCs), referred to as strains in the main text, based on core genome diversity using hierarchical Bayesian Approximation of Population Structure (hierBAPS) [46]. Secondary strain clustering (e.g., A/B/C) was assigned using the second level clustering provided by hierBAPS analysis.

A previously described carriage dataset of pneumococcal isolates from Massachusetts, US was also used to explore NFDS dynamics. For our analysis, we used the original population stratification of 16 strains identified by Croucher and colleagues [28,40]. For both datasets, we then classified strains by serotype composition as VT, NVT, or mixed (VT-NVT). The methods for whole-genome sequencing and genome assembly, and population genomic analysis have been described elsewhere [21].

For the present analysis, we focused on 937 pneumococcal carriage isolates from the Southwest US collected during 3 study periods (referred to as epochs E1–E3 in [21]): pre-vaccine–population equilibrium (E1, 1998–2001); peri-PCV7–population perturbation (E2, 2006–2008); post-PCV7–population equilibration (E3, 2010–2012) (S4 Fig). The pre-vaccine period preceded the introduction of PCV7, whereas peri- (i.e., perturbation) and post- provided snapshots 5–6 and 10–12 years, respectively, after the introduction of PCV7. Although the post-vaccine period includes, in part, the introduction of PCV13, we have previously shown that the majority of the sample was obtained when the impact of PCV13 was minimal [21]. This is supported by a sensitivity analysis to assess the effect of including all post-vaccine isolates by splitting sample into pre- and post-vaccine and testing independently (2010 versus 2011–2012).

For the additional dataset of carriage isolates from Massachusetts, we considered 133 isolates collected in 2001 as E1—the initial equilibrium population, even though the PCV7 vaccine was introduced in these communities in 2000 and therefore would be "peri-vaccine–population perturbation" in terms of the timing of vaccine introduction. We then considered 280 strains collected in 2007 as E3—the population we were predicting, which again fell into the window of "peri-vaccine–population perturbation" based on timing of vaccine introduction (S4 Fig). Comparatively, the elapsed vaccine time differed considerably, being 6 years in the Massachusetts sample compared with 10–12 years in the southwest US sample. Based on previous analysis of the Southwest US data, accessory gene frequencies were still experiencing perturbation 5–6 years after vaccine introduction (See S3 Fig in [21]). Therefore, it is likely that the Massachusetts sample had not yet reached a new post-vaccine equilibrium. We considered serotypes 4, 6A, 6B, 6E, 9V, 14, 18C, 19F, and 23F as PCV7 vaccine type. For each strain, we computed the proportion of PCV7 VT and NVT. Three serogroup 6 serotypes were included because it has previously been shown that the serotype 6B component of PCV7 was cross-protective against 6A and that 6E produces a 6B capsular polysaccharide [59]. Further, cross-reactivity is consistent with the observed elimination of 6A and 6E in the study population after the introduction of PCV7 [21].

The observed changes in prevalence were estimated as $x_i^{post} - x_i^{pre}$, where $x_i^{post}$ is the prevalence of strain *i* post-vaccine, and $x_i^1$ is the prevalence of strain *i* pre-vaccine. As a null model for vaccine impact (pro rata model), we calculated the expected prevalence for each strain if its VT representation declined to zero in the whole population from pre- to post- vaccine, and its NVT representation increased proportionately to that in the whole population, and where the new NVT prevalence values $g_i$ are renormalized to sum to one. We defined the prevalence change as $x_i^3 - g_i$. To determine significant deviations of the observed post-vaccine strain prevalence from the pro rata model, we sampled 10,000 bootstrap replicates with replacement from the E1 sample and calculated the pro rata prevalence changes for each replicate. We then plotted the 2.5%, 50%, and 97.5% quantiles of these resampled predictions in Fig 1B. We defined $x_i^3$ as significantly different from the null expectation if the strain's prevalence change was outside the central 95% of the bootstrap distribution of the predicted value.

## Pneumococcal pangenome analysis

As previously described, pangenome analysis of 937 taxa was carried out using Roary v3.12.0 (https://github.com/sanger-pathogens/Roary) [21]. The resulting presence/absence matrix was used to generate a binary accessory genome alignment of 2,371 clusters of orthologous groups (COGs). This binary alignment was used to infer a maximum likelihood (ML) phylogeny using RAxML v8.2 (https://github.com/stamatak/standard-RAxML) with BINGAMMA substitution model and 100 bootstrap replicates [60]. The same approach was used to infer a ML

phylogeny of SNPs found in the core genome using the GTR-GAMMA substitution model. Serotype, collection period (epoch), and strain (SC) assignment were visualized in relation to the accessory genome phylogeny. We then imported the phylogeny into R (https://www.r-project.org/) using APE v4.1 and computed the mean pairwise patristic distance among all strains using the *meandist* function in the R package Vegan v2.4–67 [61]. Hierarchical clustering of scaled between-strain patristic distances was visualized using *heatmap.2* in ggplots v3.0.1. Last, core and accessory genome divergence was compared with the absolute fitness difference among strains. For the additional carriage dataset from Massachusetts, the presence/absence matrix was obtained from the online repository available at https://www.nature.com/articles/sdata201558.

## Predicted fitness

In the Southwest US dataset, we identified 35 strains among 937 isolates. This included a polyphyletic grouping of strains present at low frequencies in the overall population (SC-27). Pre-vaccine, 2 strains (SC-10 and SC-24) were not sampled, having only been observed after the introduction of vaccine. Further, 2 strains (SC-22 and SC-23) had no NVT component pre-vaccine but did post-vaccine. For these 4 strains, we imputed pre-vaccine accessory gene frequencies by subsampling representative taxa from the first time point when they were observed (peri-vaccine period in both instances). This allowed us to calculate the fitness of these strains. Three additional strains (SC-04C, SC-12, and SC-17) were excluded because they had no NVT isolates present pre-vaccine or were not observed post-vaccine (i.e., they were comprised solely of VT isolates); therefore, their fitness could not be imputed nor their prevalence change. Finally, there were a few instances of strains that contained both VT and NVT serotypes. Where this was the case, for the purposes of considering the NVT portion of such strains, we removed the VTs and considered the remainder in isolation as an NVT strain. This was repeated for 14 of 16 strains in the carriage dataset from Massachusetts. This required imputing 5 strains that were not sampled pre-vaccine. A detailed list of datasets and strains included in each analysis is provided in S2 Table.

For the 2 previously unobserved strains (SC-10 and SC-24) in the primary dataset, we assessed the degree to which their accessory genome composition may have contributed to emergence after the introduction of PCV7 by comparing their fitness with strains found in the Massachusetts dataset [28,40]. To do this, we repeated the pangenome analysis using a merged dataset of 1,554 carriage isolates (including all genomes from [24]). Population structure (determination of strains) of the combined sample was assessed with hierBAPS (https://github.com/gtonkinhill/rhierbaps), and accessory gene filtering was conducted as previously detailed. Frequencies of accessory genes were determined for each strain in the Massachusetts dataset, and the predicted fitness values were calculated by comparing those frequencies to $(e − f)$ in the primary Southwest US dataset. The distribution of fitness values in the Massachusetts dataset were assessed and compared with the 2 emergent strains to determine their ranking. Last, to predict the impact of PCV13 on the pneumococcal population, we repeated the quadratic programming analysis on the post-vaccine population. To do this, we recalculated the change in strain prevalence resulting from the removal of 6 additional PCV13 VT serotypes (1, 3, 5, 6A, 7F, 19A) and determined the predicted fitness for each extant NVT strain to identify those with positive values, i.e., those that will likely be more successful in the PCV13 era.

## Post-vaccine equilibrium frequencies via quadratic programming

Using 2,371 accessory genes present in 5%–95% of taxa of the Southwest US dataset, we determined pre-vaccine accessory gene frequencies for each strain, considering NVT taxa only. For

this, we focused on 27 major strains which (1) had NVT taxa present pre-vaccine and (2) were not polyphyletic. This excluded 8 strains (SC-04C, SC-10, SC-12, SC-17, SC-22, SC-23, SC-24, and SC-27) and replicated what would have been possible with the available pre-vaccine data. S5 Fig shows the distribution of the 2,371 accessory genes among isolates belonging to the 27 strains. This figure was also used to test the assumption that the impact of recombination on the accessory genome is negligible over our study period, where we compared the pre-vaccine and post-vaccine accessory gene frequencies for each NVT strain. For the 27 strains, we computed the predicted prevalence of each strain such that post-vaccine accessory gene frequencies approached as closely as possible to pre-vaccine frequencies by using a quadratic programming approach. Quadratic programming involves optimizing a quadratic function based on several linearly constrained variables [62] and was done using the package quadprog v1.5–5 implemented in Rstudio v1.0.143 with R v3.3.19 [63]. Details of this implementation can be found in the R code provided. This was then repeated using (1) 17,101 biallelic polymorphic sites found in 1,111 genes in the core genome and present among 5%–95% of taxa and (2) 5,853 biallelic polymorphic sites found in 272 metabolic genes present in the core genome and present among 5%–95% of taxa. We then conducted a sensitivity analysis using genes present in 1%–99% and 2.5%–97.5% of taxa and found the results did not differ significantly from those obtained using genes present among 5%–95% of taxa. Detailed methods for the ascertainment of genomic loci are in the work by Azarian and colleagues [21].

Using 1,056 accessory genes present in 5%–95% of taxa of the Massachusetts dataset, we determined pre-vaccine accessory gene frequencies for each strain, considering NVT taxa only. For this, we focused on 9 major strains, which had NVT taxa present pre-vaccine and were not polyphyletic (SC-1, SC-2, SC-4, SC-8, SC-9, SC-10, SC-11, SC-12, SC-16). This excluded 7 strains and replicated what would have been possible with the available pre-vaccine data. For the 9 strains, we computed the predicted prevalence of each strain such that post-vaccine accessory gene frequencies approached as closely as possible to pre-vaccine frequencies using a quadratic programming as described here previously.

For each model, we evaluated accuracy by determining if the slope and intercept of the predicted and observed strain frequencies were close to one and zero, respectively. Goodness-of-fit statistics including sum of squares due to error (SSE), root mean squared error (RMSE), and degrees of freedom adjusted $R$-squared (Adj. $R^2$) were used to evaluate each model. In addition to assessing how well we could predict post-PCV7 prevalence, we also tested if we accurately inferred whether a strain would increase or decrease after the introduction of vaccine. To do this, we calculated the observed prevalence trajectory from pre- to post-vaccine and compared that with the predicted trajectory, identifying those with significantly positive or negative risk differences using Fisher's exact test.

### Ethics statement

The Navajo Nation, White Mountain Apache tribe, and the IRBs of the Johns Hopkins Bloomberg School of Public Health, the Navajo Nation, and the Phoenix Area IHS approved this study.

### Supporting information

**S1 Fig. Maximum likelihood phylogeny inferred from an alignment of single-nucleotide polymorphisms present in the core genome of 937 isolates obtained from a subset of 3 studies of pneumococcal carriage conducted among in the Southwest US from 1998 to 2012.** The collection epoch pre-vaccine (E1) and peri/post-vaccine (E2+E3) are indicated by the gray and orange bars, respectively. Thirty-five strains (i.e., SCs), which were identified by

subdividing the pneumococcal population on core genome diversity, are shaded by color on the phylogeny and labeled to the right. Sub-clusters (e.g., A/B/C) were identified using the second-order clustering of BAPS analysis. The phylogeny is ordered by branch decreasing branch length, and strains are numbered such that closer numbers are more genomically similar. The text color of the strain label indicates the serotype composition. The heatmap indicates PCV7 VTs and NVTs and collection period, illustrating the removal of vaccine serotypes by the introduction of PCV7. As described in the text, 2 NVT strains were not present pre-vaccine but subsequently increased thereafter. See S2, S3 Data and S1 Code for details. NVT, nonvaccine serotype; PCV-7, 7-valent pneumococcal conjugate vaccine; SC, sequence cluster; VT, vaccine serotype.

(EPS)

**S2 Fig. Analysis of 616 pneumococcal carriage isolates collected from Massachusetts dataset. A**, Peri-vaccine–E1 (2001) to Peri-vaccine–E3 (2007) change in prevalence of 16 strains (sequence clusters, SCs). Strains are ordered from highest to lowest pre-vaccine prevalence. **B**, Change in prevalence from Peri-vaccine–E1 to Peri-vaccine–E3 ordered by strain as in (**A**). Observed changes in prevalence are represented by points colored by serotype composition of the strain: nonvaccine type, PCV7 vaccine type, and mixed (vaccine and nonvaccine types). The point and whiskers show the prevalence change expected if all VT strains were removed and NVT increased pro rata to their pre-vaccine prevalence. The dot is the median, and the whiskers give the 2.5% and 97.5% quantiles of predicted changes under the null model using 10,000 bootstraps from pre-vaccine and post-vaccine samples. Significant differences are denoted with plus and minus signs specifying strains that were significantly more ($n = 5$) or less ($n = 2$) common, respectively, than expected under the null model. **C**, Scatterplot of observed versus predicted prevalence of 9 strains at post-vaccine equilibrium based on quadratic programming. These 9 strains contained at least one NVT strain pre-vaccine. Points are colored based on serotype composition as described in panel **A**. Perfect predictions would lie on the dotted line of equality (1:1 line). The shaded grey region shows the confidence interval from the linear regression model used to test for deviation of the observed versus predicted values compared to the 1:1 line. **D–E**, Comparison of the predicted prevalence change from quadratic programming analysis using accessory genes (**D**, $p = 0.38$; intercept 95% CI −0.08 to 0.03; slope 95% CI −0.24 to 1.32) and naive pro rata model (**E**, $p = 0.07$; intercept 95% CI −0.11 to 0.02; slope 95% CI −0.65 to 1.03) as shown in panel **A**. Goodness-of-fit statistics including SSE, RMSE, and degrees of freedom Adj. $R^2$ are given for each model. The lower SSE indicates a better model fit. Panel **A** presents the frequencies of 16 strains based on the entire population, whereas the analysis for **C**, **D**, and **E** necessitated the use of frequencies based on 9 strains (see S2 Table for more details). See S1 Data and S1 Code for details. Adj. $R^2$, adjusted *R*-squared; NVT, nonvaccine serotype; PCV-7, 7-valent pneumococcal conjugate vaccine; RMSE, root mean squared error; SC, sequence cluster; SSE, sum of squares due to error; VT, vaccine serotype.

(TIFF)

**S3 Fig. Core and accessory genome divergence of the Southwest US dataset. A**, Comparison of core and accessory genome divergence, quantified as the mean patristic distance between strains (SCs) on respective core and accessory genome maximum likelihood phylogenies. Each point represents a pairwise comparison of the mean between-strain patristic distances. Points are colored by the vaccine type (VT, NVT, and mixed) of strains in the pairwise comparison. Pairwise comparisons between strain sub-clusters (e.g., SC-25A and SC-25B, see S1 Fig phylogeny) and between the unencapsulated nontypeable SC-01 and other strains have been excluded. This removes the most and least divergent pairwise comparisons, which otherwise

artificially inflated the linear relationship between core and accessory genome distances. The dotted line represents the fit of a linear regression model to the data with 95% CI shaded in gray. Contour lines show 2-dimensional kernel density estimation based on the distribution of points. **B**, Core genome divergence (as defined here based on the core genomic phylogeny patristic distances) and the absolute fitness difference among 31 strains presented in Fig 3A. **C**, Accessory of core genome divergence (as defined here based on the core genomic phylogeny patristic distances) and the value of the fitness difference among 31 strains presented in Fig 3A. Of note, there is a considerable range in predicted fitness difference among strains that have similar accessory and core genome divergence (e.g., core genome divergence of 0.1 or accessory genome divergence of 1.0–1.5). See S1 Data and S1 Code for details. NVT, nonvaccine serotype; SC, sequence cluster; VT, vaccine serotype.
(TIFF)

**S4 Fig. Comparison of sampling periods for Southwest US and Massachusetts datasets.** In the Southwest US dataset, the pre-vaccine sample (*n* = 274) was collected between March 1998 and April 2001 from the well-defined control communities of the group-randomized trial of the PCV7 vaccine (i.e., no community level vaccine exposure). The subsequent sample was collected from 2006–2008 during a PCV7 impact study. We refer to this time point as the population perturbation or "peri-PCV7" period because previous analysis of these data has shown that the pneumococcal population was destabilized. The PCV13 vaccine was later introduced in 2010; however, our sample was collected prior to vaccine coverage reaching 80% among children <5 and sensitivity analysis detailed in the main text illustrated the limited impact of PCV13 during this time period. For the Massachusetts carriage dataset, we considered 133 isolates collected in 2001 as "Peri-vaccine–E1," and 280 strains collected in 2007 as "Peri-vaccine–E3," given that the PCV7 vaccine was introduced in these communities in 2000. As noted in the main text, this sample was the most optimal for comparison considering its limitations of population genomic samples of *S. pneumoniae*. NVT, nonvaccine serotype; PCV-7, 7-valent pneumococcal conjugate vaccine; VT, vaccine serotype.
(EPS)

**S5 Fig. Distribution of clusters of orthologous groups among 27 NVT strains that were observed Southwest US sample pre-vaccine, colored by the epoch.** Histograms represent the presence and absence of accessory genes that were present in between 5% and 95% of the entire sample. Note, each strain will have a variable number of accessory genes depending on their accessory genome diversity. Within each strain, the distribution of accessory genes remained relatively stable from pre- to post-vaccine. See S1 Data and S1 Code for details. NVT, nonvaccine serotype.
(TIFF)

**S1 Table. Strain counts and prevalence by period and type (VT, NVT) for the Southwest US dataset.** NVT, nonvaccine serotype; VT, vaccine serotype.
(DOCX)

**S2 Table. Description of dataset and strains used in each analysis.** For each figure, we have listed the analysis, dataset used (SW US or Mass), sample used, strains included, and explanation of any strain exclusions. Mass, Massachusetts; SW US, Southwest US.
(DOCX)

**S1 Code. Supplemental Code.** Rmd and pdf files for all code used replicate each analysis and reproduce main and supporting figures. Source data is provided in supporting data.
(ZIP)

**S1 Data. Data for reproducing all main and supporting figures.**
(XLSX)

**S2 Data. Phylogenetic tree file used to create S1 Fig.**
(TRE)

**S3 Data. Metadata file used to create S1 Fig.**
(CSV)

## Acknowledgments

The ideas in this paper and the collaborative relationships underlying them are owed in significant part to the Permafrost Conference on Microbial Genomics, of Blessed Memory.

## Author Contributions

**Conceptualization:** Taj Azarian, Brian J. Arnold, Laura L. Hammitt, Raymond Reid, Mathuram Santosham, Robert C. Weatherholtz, Katherine L. O'Brien, Marc Lipsitch, William P. Hanage.

**Data curation:** Pamela P. Martinez, Xueting Qiu, Marc Lipsitch.

**Formal analysis:** Taj Azarian, Pamela P. Martinez, Brian J. Arnold, Jukka Corander, Nicholas J. Croucher, Marc Lipsitch.

**Funding acquisition:** Marc Lipsitch, William P. Hanage.

**Investigation:** Taj Azarian, Pamela P. Martinez, Xueting Qiu, Lindsay R. Grant, Jukka Corander, Christophe Fraser, Nicholas J. Croucher, Laura L. Hammitt, Raymond Reid, Mathuram Santosham, Robert C. Weatherholtz, Stephen D. Bentley, Katherine L. O'Brien, Marc Lipsitch, William P. Hanage.

**Methodology:** Taj Azarian, Pamela P. Martinez, Lindsay R. Grant, Jukka Corander, Christophe Fraser, Nicholas J. Croucher, Laura L. Hammitt, Raymond Reid, Mathuram Santosham, Robert C. Weatherholtz, Stephen D. Bentley, Katherine L. O'Brien, Marc Lipsitch, William P. Hanage.

**Project administration:** Robert C. Weatherholtz, Marc Lipsitch, William P. Hanage.

**Resources:** Pamela P. Martinez, Lindsay R. Grant, Laura L. Hammitt, Raymond Reid, Mathuram Santosham, Katherine L. O'Brien, Marc Lipsitch.

**Software:** Pamela P. Martinez.

**Supervision:** Jukka Corander, Laura L. Hammitt, Raymond Reid, Robert C. Weatherholtz, Katherine L. O'Brien, Marc Lipsitch, William P. Hanage.

**Validation:** Taj Azarian, Pamela P. Martinez, Xueting Qiu, Christophe Fraser, Robert C. Weatherholtz, Katherine L. O'Brien, William P. Hanage.

**Visualization:** Taj Azarian, Pamela P. Martinez, Brian J. Arnold.

**Writing – original draft:** Taj Azarian, Pamela P. Martinez, Marc Lipsitch, William P. Hanage.

**Writing – review & editing:** Taj Azarian, Pamela P. Martinez, Brian J. Arnold, Xueting Qiu, Lindsay R. Grant, Jukka Corander, Christophe Fraser, Nicholas J. Croucher, Laura L. Hammitt, Raymond Reid, Mathuram Santosham, Robert C. Weatherholtz, Stephen D. Bentley, Katherine L. O'Brien, Marc Lipsitch, William P. Hanage.

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
