## [Editor Report · Decision Letter 0]

16 Mar 2020

Dear Dr Azarian, 

Thank you for submitting your manuscript entitled "Frequency-dependent selection can forecast evolution in Streptococcus pneumoniae" for consideration as a Research Article by PLOS Biology.

Your manuscript has now been evaluated by the PLOS Biology editorial staff, as well as by an academic editor with relevant expertise, and I'm writing to let you know that we would like to send your submission out for external peer review.

Please re-submit your manuscript within two working days, i.e. by Mar 18 2020 11:59PM.

Kind regards,

Roli Roberts

Senior Editor

PLOS Biology

---

## [Decision Letter · Decision Letter 1]

12 May 2020

Dear Dr Azarian,

Thank you very much for submitting your manuscript "Frequency-dependent selection can forecast evolution in Streptococcus pneumoniae" for consideration as a Research Article at PLOS Biology. Your manuscript has been evaluated by the PLOS Biology editors, an Academic Editor with relevant expertise, and by three independent reviewers.

You'll see that the reviewers are broadly positive about your study, but each raises some concerns that will need addressing, or makes requests for additional information. In light of the reviews (below), we will not be able to accept the current version of the manuscript, but we would welcome re-submission of a much-revised version that takes into account the reviewers' comments. We cannot make any decision about publication until we have seen the revised manuscript and your response to the reviewers' comments. Your revised manuscript is also likely to be sent for further evaluation by the reviewers.

We expect to receive your revised manuscript within 2 months. 

**IMPORTANT - SUBMITTING YOUR REVISION**

*Re-submission Checklist*

*Published Peer Review*

*PLOS Data Policy*

*Blot and Gel Data Policy*

Sincerely,

Roli Roberts

Senior Editor

PLOS Biology

REVIEWERS' COMMENTS:

Reviewer #1:

[identifies himself as Chris Illingworth]

This manuscript describes the use of a method to predict strain frequencies in S. pneumoniae populations. The central principle is that vaccination induces a perturbation to the population; strains which are most similar to the collective set of what has been removed are at an advantage, and grow in frequency, the population restoring its initial genetic balance. It is a very interesting piece of work and is generally well presented.

Major comments:

While the results shown are promising, my concern with this work is the extent to which the data are representative of S. pneumoniae populations in general. The evolution of a population comprises both changes arising from standing variation (e.g. changes in the frequencies of existing variants or strains in the population), and changes arising from de novo events (including the onset of new mutations, migration into the local environment, and so forth). In common with most approaches to evolutionary prediction, the model presented considers only the first kind of evolutionary change.

In the dataset shown, the initial population comprised 33 strains (Fig. 1), and 12 years later the same strains predominated with the loss only of two strains that were vaccinated against and the gain of only two more strains. The S. pneumoniae population considered is therefore something of a closed system.

This leads to questions about the data used for the study.

i) To what extent are these data representative of S. pneumoniae populations in general? It is noted that the data were collected from Native American populations. Could these represent unusually closed populations in terms of the potential introduction of new strains from outside of the community? Would the same be true of bacterial populations in a major city? In general, how stable is a population against invasion on a timescale of 12 years?

ii) Why were these specific datasets chosen? It is mentioned that the second dataset is limited in the span between samples. Very many S. pneumoniae genomes have been sequenced; was the choice based simply on a lack of availability of data collected before and after a specific vaccination program?

iii) It is noted that a subset of the data from three studies was taken. Please note how this subset was chosen.

My view is that the method presented here is of interest even if it is limited in application to unusually closed systems; what I am unsure of is the generality of its predictive value. Further comment should be made on the data.

Minor comments:

1. There is an ambiguity regarding the composition of the individual strains. Line 153 mentions the frequency of a gene within a single strain. Is it the case that this frequency is predominantly either zero or one? If not, the model potentially implies that within-strain evolution would be expected in addition to the overall changes in strain frequency. Was this looked at?

2. In equation 1 I think that x_i is the frequency of strain i after the population has been rescaled to account for the removal of the vaccinated-against strains (i.e. the frequencies from the pro-rata model). This should be clarified.

3. Would a combination of information from the combined and accessory genomes provide a better result than taking each set individually? 

4. In Figure S3 the fittest genotypes according to the accessory genome measure are significantly more diverged than the bulk of the population; is there a reason for this? To what extent does the divergence measure shown differ from the mean divergence from a vaccinated strain?

5. Figure 1 is nice in showing the changes in different strains relative to the pro rata model; could a similar figure be created for the fitness-based model?

6. The colours in Figure 1 were too similar for my liking. The contrast between blue and red is clear, but the purple is hard to distinguish from the blue, particularly given the two-tone nature of the chart. Please make this more distinct.

7. "Enables to predict" in the abstract - please correct the grammar here.

Reviewer #2:

[identifies himself as Kevin Bakker]

Overall, this is an interesting study, however the authors need to be a bit more clear in which samples were used for each analysis and explain the impact of the work. Figure 1 explains there are 35 strains in the study out of ~90 circulating, figure 2 randomly selects 35/90 strains, Figure 3A uses only 31 and it's not clear which of the 31 are used, and in Figs 3B-3D there are only 27 strains - it's not clear which 27 these are. There are multiple time scales discussed (6 years not long enough for evolution but 12 years is ok, 2001-2004 was a peri-vaccine for MA but 2006-2008 in the SW, whereas 2007 was post-vaccine in MA but 2010-2012 in the SW). Meanwhile in the results and discussion the SW dataset is partitioned into pre- and post-vaccine. My point being there are too many time period definitions throughout and it's extremely difficult to follow. While interesting, the authors fail to note why this work is impactful or significant. Are certain strains more pathogenic? What proportion of people are healthy/asymptomatic carriers? Is there any point to vaccination if rare NVT strains are just going to take the place of VT strains post-vac? Do we need new vaccines (e.g. each booster dose has a different combination of strains)? Will immigration of VT strains from unvaccinated individuals re-establish dominance post-vaccination? What is the impact of the polysaccharide vaccine on strain prevalence? Since the PCV is only administered to children and polysaccharide vaccine to seniors, how do middle aged individuals contribute to strain diversity? Essentially, why is forecasting the evolution of Streptococcus pneumoniae important? This is briefly touched on in a hypothetical manner in the discussion (lines 325-328), but per Plos Bio reviewer guidelines it needs significant work on explaining the 'novelty and significance of the findings'. 

General comments:

- I'm not familiar with PCV's, does the vaccine strain composition change over time? Or did it change over your study period, 1998-2012? (not PCV7 to PCV13, but whether the strains differed)

- Is the PCV similar to the polysaccharide vaccine? Are they composed of the same strains? I assume all these samples were collected in children, as the polysaccharide vaccine is used in adults?

- line 123 - are these 35 strains representative of strains in the US/world? The authors mentioned 90+, but what is the typical composition, are these 35 the most common?

- line 123/Fig 1 - From my understanding, the PCV covers 8/35 strains present? 

- Figure 1 - Is the post-vaccine 04C likely due to vaccine failure?

- Figure 1 - Thoughts on why strain 23 increased in prevalence? 

- Figure 1 - Why did the null model only find significant difference in SC-22 and SC-23, but not SC-21? All were mixed with little change pre- and post-vac.

- Figure 1/S1 - Were the 3 previous studies performed on the entire population? Was the entire population vaccinated? 

- line 130 - Shouldn't this be 14 years, not 12 years afterward? 1998-2012?

- line 130 and last line of caption on Fig 1 - 13/33, because you didn't include SC-12 and SC-24?

-line 701/702 - How has relative prevalence not change pre-vaccine to predicted fitness? The two bottom blue lines seem to have swapped positions?

- Fig 2A - Why are there units of time on the x-axis? Since this is conceptual, wouldn't it be more sensible to label each period as an epoch (establishing equilibrium, pre-vac equil/steady state, vacc intro, pred fitness, est. equilibrium, post vac equil/steady state)?

- line 705 - any motivation by choosing 10 simulations? These look to be fairly noisy though the general result can be seen. Why not choose the 35 strains you have in your study, rather than 35 at random?

- Fig 2B - Thoughts on why the model never incorrectly predicted an increase in fitness, but was only incorrect when predicting a decrease? Going back to Fig 1A, there are a few examples of this in the data where I would have expected to see an increase, yet there was a decrease (SC-06A, SC-06B, SC-19A, SC-07, SC-03B etc.)

- line 179/Fig 3 - Why only test 31 strains here, rather than the 35 in all other examples? Are the 4 taken out the ones either not present pre- or post-vaccine? I think this is explained on line 718, but I can only see two strains (SC-10 and SC-24) without NVT isolates before vaccination, not 4.

- line 190 - There are 33 strains in Fig 1 pre-vaccine, why only 27? Caption on Fig 3 says post-vaccine, but there are 33 of those in Fig 1...

- Fig 3 - I'm a bit confused on strain selection. In Fig 1 there are 5 mixed composition strains all present pre- and post-vaccine. In Fig 3A there are 4 and 3B-3D there are only 2.

- Even after reading the methods, the number of strains in each figure is still confusing.

- Fig 3A - Possible to include a legend indicating mixed vs NVT?

- Fig 3A - possible to move label SC-03B to whitespace to improve legibility?

- line 723 'increase'

- line 728 - any reason to choose the 1.5x IQR for annotation?

- line 730 slope, CI, etc. all with or without sample SC-09?

- line 206 - when was the first year of vaccination? After reading the methods, I see this, but it would be nice to have basic year information in the text.

- line 208 - mentions 6 years is not enough time for evolution, whereas 12 is - evidence/citation? (more on this below) From the x-axis on Fig 2A it seems like 6 years would be plenty (if those are years) - another reason to remove time units from 2A.

- line 213 - PCV13 was introduced in 2010, but lines 357-358 identify post-pcv7 as 2010-2012 which you say is 'post-vaccine sampling'.

- line 313 - I think immigration is important here. You were able to show massive difference pre- and post-vaccine, but with re-introduction of VT strains (e.g. SC-17 and SC-12) through individual movement/transmission, will they increase in prevalence again? 

- line 235 - is there already work on invasive capacity of various strains?

- line 328 - are there less-virulent strains?

- line 328 - even if we selected for drug susceptible lineages, wouldn't non-susceptible lineages pop up (per your results)?

- line 371 - gene frequencies are dynamic by nature, especially in small sample sizes - are there time series showing stable gene frequencies in small populations without vaccination? Fig S3 from the plos path paper is as likely to be noise as it is perturbation on its way to equilibrium. 

Reviewer #3:

This study aims to tackle the very difficult task of predicting how pathogen populations will change over time using S. pneumoniae as a study system. The authors use frequencies of accessory genes to predict changes in the pneumococcal population after vaccination, hypothesising that these frequencies reflect negative frequency-dependent selection (NFDS) on the gene products.

I very much enjoyed the study and found the methods robust and creative. There are few things that I would have liked to see added:

1. Even though the authors say there is not much difference in terms of geography, I would like to see that in a plot and some statistics confirming that. My gut feeling is that immigration will pay a role in diversity.

2. Also, I would like to see discussion on redundancy of some of these accessory genes and how that might affect predictive power.

3. Finally, a paragraph on recombination and how that affects lineage composition would be appropriate.

---

## [Decision Letter · Decision Letter 2]

5 Aug 2020

Dear Dr Azarian,

Thank you for submitting your revised Research Article entitled "Frequency-dependent selection can forecast evolution in Streptococcus pneumoniae" for publication in PLOS Biology. I've now obtained advice from two of the original reviewers and have discussed their comments with the Academic Editor. 

Based on the reviews, we will probably accept this manuscript for publication, assuming that you will modify the manuscript to address the remaining points raised by the reviewers. IMPORTANT: Please also make sure to address the Data Policy-related requests noted at the end of this email.

We expect to receive your revised manuscript within two weeks. Your revisions should address the specific points made by each reviewer. In addition to the remaining revisions and before we will be able to formally accept your manuscript and consider it "in press", we also need to ensure that your article conforms to our guidelines. A member of our team will be in touch shortly with a set of requests. As we can't proceed until these requirements are met, your swift response will help prevent delays to publication.

*Copyediting*

*Published Peer Review History*

*Early Version*

*Submitting Your Revision*

Sincerely,

Roli Roberts

Senior Editor,

rroberts@plos.org,

PLOS Biology

DATA POLICY:

Many thanks for flagging the location of the raw data in BioProject, and the code and phylogenies in GitHub. However, we also ask that all individual quantitative observations that underlie the data summarized in the figures and results of your paper be made available in one of the following forms:

Regardless of the method selected, please ensure that you provide the individual numerical values that underlie the summary data displayed in the following figure panels as they are essential for readers to assess your analysis and to reproduce it: Figs 1AB, 2B, 3ABD, S1, S2ABCDE, S3ABC, S5. NOTE: the numerical data provided should include all replicates AND the way in which the plotted mean and errors were derived (it should not present only the mean/average values).

REVIEWERS' COMMENTS:

Reviewer #1:

[identifies himself as Chris Illingworth]

I am happy with the response made to previous comments. For clarity I suggest in line 417ff. using the notation x_i^{post} and x_i^{pre} rather than 3 and 1.

Reviewer #2:

[identifies himself as Kevin Bakker]

This version of the manuscript is much improved. The additional text and figures (Particularly Fig S4, Table S2, and small changes/additions to the methods and discussion) helped my understanding of the samples, periods, and motivation. The authors addressed all of my major concerns and I only have a minor comment below.

-Not sure if it was intentional, but the naming of supplementary figures seemed odd (e.g. 'S2 Figure' and 'Figure 1 and S1 Figure' rather than just 'Figure S2' or 'Figures 1 and S1').

---

## [Editor Report · Decision Letter 3]

18 Sep 2020

Dear Dr Azarian,

On behalf of my colleagues and the Academic Editor, J. Arjan G. M. de Visser, I am pleased to inform you that we will be delighted to publish your Research Article in PLOS Biology. 

Early Version

PRESS 

Kind regards,

Vita Usova 

Publication Assistant, 

PLOS Biology

on behalf of

Roland Roberts,

Senior Editor

PLOS Biology